# Editing of the TRIM5 Gene Decreases the Permissiveness of Human T Lymphocytic Cells to HIV-1

**DOI:** 10.3390/v13010024

**Published:** 2020-12-25

**Authors:** Kevin Désaulniers, Levine Ortiz, Caroline Dufour, Alix Claudel, Mélodie B. Plourde, Natacha Merindol, Lionel Berthoux

**Affiliations:** Department of Medical Biology, Université du Québec à Trois-Rivières, Trois-Rivières, QC G9A 5H7, Canada; kevin.desaulniers2@uqtr.ca (K.D.); levine_muse@hotmail.com (L.O.); caroline.dufour@uqtr.ca (C.D.); alix.claudel@yahoo.com (A.C.); melodie.bplourde@uqtr.ca (M.B.P.); natacha.merindol@uqtr.ca (N.M.)

**Keywords:** HIV-1, TRIM5α, genome editing, CRISPR, restriction factor, interferon

## Abstract

Tripartite-motif-containing protein 5 isoform α (TRIM5α) is a cytoplasmic antiretroviral effector upregulated by type I interferons (IFN-I). We previously showed that two points mutations, R332G/R335G, in the retroviral capsid-binding region confer human TRIM5α the capacity to target and strongly restrict HIV-1 upon overexpression of the mutated protein. Here, we used clustered regularly interspaced short palindromic repeats (CRISPR)-Cas9-mediated homology-directed repair (HDR) to introduce these two mutations in the endogenous human TRIM5 gene. We found 6 out of 47 isolated cell clones containing at least one HDR-edited allele. One clone (clone 6) had both alleles containing R332G, but only one of the two alleles containing R335G. Upon challenge with an HIV-1 vector, clone 6 was significantly less permissive compared to unmodified cells, whereas the cell clones with monoallelic modifications were only slightly less permissive. Following interferon (IFN)-β treatment, inhibition of HIV-1 infection in clone 6 was significantly enhanced (~40-fold inhibition). TRIM5α knockdown confirmed that HIV-1 was inhibited by the edited TRIM5 gene products. Quantification of HIV-1 reverse transcription products showed that inhibition occurred through the expected mechanism. In conclusion, we demonstrate the feasibility of potently inhibiting a viral infection through the editing of innate effector genes. Our results also emphasize the importance of biallelic modification in order to reach significant levels of inhibition by TRIM5α.

## 1. Introduction

Individuals infected with the human immunodeficiency virus type 1 (HIV-1) are treated with combination antiretroviral therapy (cART). Despite successfully reducing viral loads to undetectable levels in a large fraction of the treated patients [1], drugs administered under cART have significant side effects [2], complicating adherence. Moreover, they are not curative, as they do not target latently integrated HIV-1 that constitute the main reservoir [3], and are inefficient in some anatomical sanctuaries such as the central nervous system [4]. Genetic interventions offer the potential to durably suppress HIV-1 while avoiding the need for lifelong pharmacological treatments. The HIV-1 entry co-receptor CCR5 has been the most studied target for HIV-1 gene therapy. The goal is to phenotypically mimic the CCR5∆32/∆32 genotypes of the donor used for the “Berlin patient” [5]. Zinc finger nucleases (ZFNs) have been designed to knock out the CCR5 gene, and phase I clinical trials demonstrated that patient-derived ZFN-engineered CCR5-knockout T cells were effective in controlling HIV-1 viral loads in some patients following treatment interruption [6]. Several clustered regularly interspaced short palindromic repeats (CRISPR)-based approaches have also been developed to edit the CCR5 gene [7,8,9]. However, results from a patient in whom CXCR4-tropic HIV-1 rebounded the following transplantation with CCR5∆32/∆32 cells suggest that this approach alone may not be successful for patients that harbor even very small amounts of HIV-1 that can use the CXCR4 co-receptor for entry into cells [10,11].

Retrovirus infections may be inhibited by a family of innate immune effectors, also called restriction factors [12,13]. These proteins may show some level of efficacy without external stimulation (“intrinsic immunity”), but their expression is typically stimulated by IFN-I. Restriction factors act at different stages of retroviral replication and through a variety of mechanisms (reviewed in [14]). Tripartite-motif-containing protein 5 isoform α (TRIM5α) is a cytoplasmic restriction factor encoded by the interferon-stimulated gene (ISG) *TRIM5* [15,16] (reviewed in [17]).

TRIM5α targets retroviruses shortly after their entry into the cell’s cytoplasm [18]. TRIM5α, like other members of the TRIM protein family, has RING, B-box and Coiled-coil domains at its N-terminus [19]. The RING domain-associated ubiquitin ligase activity is instrumental in the restriction mechanism, as it directs some viral components as well as TRIM5α itself to proteasomal degradation [20,21]. The RING domain also promotes the formation of K63-linked ubiquitin chains that play a role in the restriction process [22,23] and activate innate immune pathways mediated by NF-κB, AP-1 and IFN-I [24,25,26,27]. At its C-terminus, TRIM5α has a SPRY/B30.2 domain whose sequence determines retroviral target specificity, i.e., which viruses are inhibited [28,29]. Upon intercepting incoming retroviral cores, TRIM5α binds to the capsid protein lattice that forms the outer side of the core. This, in turn, promotes the multimerization of TRIM5α, greatly enhancing the avidity of TRIM5α-capsid interactions [30,31]. The dimerization of TRIM5α and formation of higher-order multimers involve its central B-box and Coiled-coil domains [32,33]. As a result, the viral core is destabilized and undergoes premature disassembly [34,35,36,37], which disrupts reverse transcription of the viral genome [38,39,40]. The restriction mechanism also includes the sequestration of viral particles in cytoplasmic bodies [41,42].

Restriction by TRIM5α is species-specific, cell-type-specific and virus-specific [43,44,45]. Some HIV-1 strains are significantly restricted by human TRIM5α (huTRIM5α), in particular in HLA: B27+ or B57+ elite controller patients, because of evolutionary pressure from cytotoxic T lymphocytes in these patients results in the emergence of capsid mutants that increase sensitivity to TRIM5α [24,46,47]. However, most HIV-1 strains are poorly sensitive to huTRIM5α (<2-fold). In contrast, many nonhuman primate orthologs of TRIM5α, such as rhesus macaque TRIM5α (rhTRIM5α), restrict HIV-1 by 10- to 100-fold [48,49]. Attempts have been made to generate mutants of huTRIM5α able to efficiently target HIV-1. For instance, investigators have produced chimeric versions of huTRIM5α containing small motifs of the rhTRIM5α SPRY domain [43,50]. Other studies introduced smaller changes in the SPRY domain based on the rhTRIM5α sequence, leading to the discovery that mutations abrogating the positive charge at Arg332 increased huTRIM5α targeting of HIV-1 [51,52]. Taking a different approach, our laboratory generated huTRIM5α SPRY domain mutant libraries that were screened for their capacity to restrict HIV-1. This led us to isolate the HIV-1 inhibitory mutation R335G, among other mutations [53,54]. Furthermore, combining the R332G and R335G mutations yielded higher restriction levels compared with single mutations [53,54]. When overexpressed through retroviral transduction, R332G-R335G huTRIM5α inhibits the spread of HIV-1 by 20- to 40-fold and provides a survival advantage compared to untransduced cells [53,55].

Overexpressing TRIM5α may have detrimental consequences in vivo, as this protein is involved in processes such as inflammation [24,27] and autophagy [56]. Another caveat of lentiviral vector-mediated TRIM5α transduction is the continued expression of the endogenous, wild-type WT protein. TRIM5α proteins interact with each other, and the presence of the nonrestrictive WT protein may interfere with the antiviral activity of the restrictive mutant [57,58]. Introducing the desired mutations in *TRIM5* by gene editing represents an attractive alternative, as the therapeutic gene will be expressed at physiological levels and in an IFN-I-dependent fashion. Assuming biallelic gene editing, no WT protein would be co-expressed along with the therapeutical mutant. Previously, we transfected plasmids encoding CRISPR components (Cas9 and guide RNA, gRNA) into HEK293T cells, along with a donor DNA for HDR that bear the desired mutations [59], and obtained several cell clones containing corrected alleles [60]. However, no antiviral effect was observed, which was due to several possible reasons, including the fact that in this cell line, *TRIM5* bears a high number of mutations that may inactivate its antiviral properties (see http://hek293genome.org/v2/) [60]. In this paper, we electroporated CRISPR ribonucleoprotein (RNP) complexes and a mutation donor DNA into Jurkat T cells in order to introduce the R332G and R335G mutations. Our results demonstrate that successfully *TRIM5*-edited cells have decreased permissiveness to HIV-1 and that restriction is stimulated by IFN-I as expected.

## 2. Materials and Methods

### 2.1. Cell Culture

Jurkat T lymphocytic cells were maintained in RPMI 1640 medium (HyClone, Thermo Fisher Scientific, Ottawa, ON, Canada). HEK293T cells were maintained in DMEM medium (HyClone). The medium was switched to RPMI at the time of virus production as the virus was used to infect cells growing in RPMI. All culture media were supplemented with 10% fetal bovine serum, penicillin/streptomycin (HyClone) and low-concentration Plasmocin (InvivoGen, San Diego, CA). The Jurkat cells stably transduced with a lentivirus overexpressing R332G-R335G TRIM5α were described before [53]. Briefly, these cells were exposed to an HIV-1 vector containing both the huTRIM5α mutant and the gene of resistance to puromycin, followed by puromycin treatment to kill untransduced cells.

### 2.2. TRIM5 Editing

The CRISPR target within *TRIM5* (5′AGATAATATATGGGGCACGA) was selected using the Zhang Lab algorithm available online at crispr.mit.edu [60]. We used the Integrated DNA Technologies (IDT, Kanata, ON, Canada) ALT^®^ CRISPR-Cas9 RNP system to edit *TRIM5*. The CRISPR RNA (crRNA) and transactivating CRISPR RNA (tracrRNA), which are the RNA components of the RNP complex, were synthesized by IDT. The previously described single-stranded donor DNA (ssDNA) that includes mutations encoding 332G and 335G as well as silent mutations [60] was also synthesized by IDT. 1 μL of the RNP complex contained 22 pmol of crRNA:tracrRNA duplex and 18 pmol of Cas9 enzyme. The electroporation enhancer was diluted in IDTE buffer at a final concentration of 10.8 μM. We electroporated 3 × 10^5^ cells with 100 pmol of the donor ssDNA, 1 μL of the RNP complex and 2 μL of electroporation enhancer. Electroporations were done in the Neon transfection system (Thermo Fisher Scientific) with the following parameters: 1323 V, 10 ms and 3 pulses. The electroporated cells were incubated at 37 °C, 5% CO_2_ for 48 h.

### 2.3. Isolation and Screening of Cell Clones

To isolate single-cell clones, we seeded 6 × 96-well plates with approximately 0.5 cells per well. For each well, we used 100 μL of RPMI 1640 medium supplemented with 12% FBS, 10% of conditioned medium, penicillin/streptomycin and Plasmocin. The plates were grown for 2–4 weeks, and we obtained 47 surviving clonal cell populations. To screen clones for HDR-mediated *TRIM5* editing, genomic DNA was extracted using 30 μL of DirectPCR lysis reagent (Viagen Biotech, Los Angeles, CA, USA) mixed with 30 μL of water containing 12 μg of proteinase K. Lysis was completed overnight at 55 °C, then the lysate was heated to 85 °C for 90 min to deactivate the proteinase K. HDR editing-specific PCR was performed on 2 μL of samples using the OneTaq polymerase (New England Biolabs, Whitby, ON, Canada) and primers T5a_mut_fwd (5′-AAATAATCTACGGGGCCGGCGGCACAG) and T5a_qPCR_rev (5′- CCAGCACATACCCCCAGGAT). PCR was performed using the following parameters: 30 s at 94 °C, 30 s at 61.5 °C, 30 s at 68 °C, 30 cycles. The reaction products were analyzed by electrophoresis on agarose gels.

### 2.4. HaeIII Screening of Edited Clones and Deep Sequencing

Genomic DNA was extracted using the EZ-10 spin column genomic DNA kit (BioBasic, Markham, ON, Canada) from the cell clones found to be positive in the first screen described above, resuspended in 50 µL and quantified with a NanoDrop spectrophotometer (Thermo Fisher Scientific). We first amplified the targeted *TRIM5* region by PCR using the OneTaq polymerase, with 5 μL of genomic DNA extraction and bar-labeled primers. Those primers, huTR5aGG_seq_FOR (5′-ACACTGACGACATGGTTCTACAATCCCTTAGCTGACCTGTTA) and huTR5aGG_seq_REV (5′-TACGGTAGCAGAGACTTGGTCTCCCCCAGGATCCAAGCAGTT) bind outside the 200 nt regions aligning with the donor ssDNA. The PCR was run for 30 cycles using the following conditions: 94 °C for 30 s, 63 °C for 30 s and 68 °C for 60 s. An aliquot of each PCR product was digested with HaeIII (New England Biolabs) at 37 °C for 60 min. Reaction products were analyzed by performing electrophoresis on agarose gels. The PCR products (undigested) were sequenced on a MiSeq apparatus at Genome Quebec (McGill University, Montreal, QC, Canada), and results were analyzed using Integrative Genomic Viewer, available online (http://software.broadinstitute.org/software/igv/).

### 2.5. Virus Production

Plasmid DNA was prepared using the Plasmid Midi kit (Qiagen, Montréal, QC, Canada). HEK293T cells were seeded into 10 cm culture dishes and transfected the next day using polyethyleneimine (PolyScience, Niles, IL, USA) with the following plasmids: pMD-G (5 μg), pCNCG (10 μg) and pCIG3-N or pCIG3-B (10 μg) to produce N-MLV_GFP_ and B-MLV_GFP_, respectively; pMD-G (5 μg) and pH IV-1_NL-GFP_ (10 μg) to produce HIV-1_NL-GFP_ [61,62]. The medium was changed 6 h after transfection, and virus-containing supernatants were harvested 24, 48 and 72 h later. Supernatants were clarified by centrifuging for 10 min at 3000 rpm then filtered through 0.45 μm filters (Bio-Rad Millex Durapore PVDF syringe filters, Thermo Fisher Scientific).

### 2.6. Viral Challenges

Infections of Jurkat cells were performed in 96-well plates seeded at 10,000 cells per well the day before. Where applicable, treatment with IFN-β (PeproTech, Rocky Hill, NJ, USA) at a final concentration of 10 ng/mL was initiated 16 h prior to infection [60]. The virus amounts used were determined so that we would obtain >0.1% infected cells (FACS analysis threshold in the conditions used) while staying below saturation concentrations (≈50% infected cells), based on preliminary titration experiments. 48 h post-infection, infected cells were fixed in 2.5% formaldehyde. The percentage of GFP-positive cells was determined using an FC500 MPL cytometer (Beckman Coulter, Brea, CA, USA) with the FCS Express 6 analysis software (De Novo Software, Pasadena, CA, USA). The percentage of GFP+ cells among total intact cells was determined by gating them after excluding cell debris.

### 2.7. Knockdowns

Cells were knocked down for TRIM5α, or luciferase as a control, by lentiviral transduction of pAPM vectors expressing miR30-based shRNAs [27], as extensively described in previous publications [24,63]. Transduced cells were treated with puromycin at 1 μg/mL for one week; all mock-transduced cells were killed in those conditions.

### 2.8. HIV-1 cDNA Quantification

Seven hundred fifty thousand cells per well were seeded in 24-well plates using 0.6 mL of medium per well, 24 h prior to infection. Where applicable, IFN-β (10 ng/mL), MG132 (Sigma-Aldrich, Oakville, ON, Canada; 1 μg/mL) and nevirapine (Sigma-Aldrich; 40 or 80 μM) were added 16 h (IFN-β) or 4 h (MG132, nevirapine) prior to infection. Infection was done at a multiplicity of infection of 0.1 with HIV-1_NL-GFP_ passed through 0.2 μM filters (MilliporeSigma Durapore) and treated with 20 U/mL DNaseI (New England Biolabs) for 60 min at 37 °C. Cells were infected for 6 h; then, total cellular DNA was extracted using the EZ-10 spin column genomic DNA kit (BioBasic) and quantified using the NanoDrop spectrophotometer. HIV-1 cDNAs were amplified in quantitative PCR (qPCR) reactions using primers specific for GFP or for the cellular gene GAPDH as a control, as described before [64]. PCR reactions were done in a final volume of 20 μL, containing 1X of SensiFast SYBR Lo-ROX kit (Bioline, Meridian Biosciences, Memphis, TN), 400 nM sense and antisense primers and between 150 to 400 ng of DNA. After 3 min of incubation at 95 °C, 40 cycles of amplification were achieved as follows: 5 s at 95 °C, 10 s at 60 °C, 15 s at 72 °C in an Agilent Mx3000P instrument. Reactions were performed in duplicate, and the threshold cycle was determined using the MxPro software (Agilent). HIV-1 cDNA levels were normalized to those of GAPDH, which was amplified simultaneously with the same PCR parameters, using the ΔCt method (cycle threshold for the sequence of interest minus cycle threshold for the control sequence). Relative HIV-1 cDNA copy numbers were then normalized to levels found in infected/untreated control (parental) cells, which were set at 100%.

## 3. Results

### 3.1. TRIM5 Editing

Previously, we had designed three gRNAs leading to Cas9-mediated DNA cuts in the vicinity of Arg332 and Arg335 in *TRIM5* [60]. gRNA1 induces the cut closest to the desired mutations, specifically just upstream to the Arg332 triplet (Figure 1), and was the one used in this study. The HDR donor ssDNA, which is antisense to the gRNA and 200 nts long with homology arms of similar size, was the same as in the previous study [60]. Its central section contains the mutations to be introduced, as represented in Figure 1. In addition to mutations substituting arginine residues into glycine at positions 332 and 335, silent mutations are introduced to prevent resection of the HDR-corrected DNA by Cas9 through changes in both the gRNA binding site and the protospacer-adjacent motif (PAM). The silent mutations also create a HaeIII restriction site to facilitate subsequent screening (Figure 1).

Jurkat cells were electroporated with CRISPR-RNP complexes along with the donor ssDNA. An aliquot of the transfected cells was subsequently analyzed for the presence of HDR-modified alleles. For this, we extracted DNA from the whole cell population and subjected it to PCR using primers designed to specifically amplify HDR-modified *TRIM5* [60]. We detected a PCR product of the expected size (157 nt long), suggesting the presence of the desired modifications (not shown). To isolate cellular clones of edited cells, we seeded the transfected cells in 96-well plates at 0.5 cells per well. We obtained 47 clonal Jurkat cell populations that were individually analyzed. An aliquot of each population was lysed and subjected to the HDR-edited-specific PCR assay. We found that 6 clones (13%) were positive in this screen: clones 6, 8, 12, 17, 30, 38 (Appendix A). We then performed a second PCR-based analysis on the six positive clones as well as eight negative ones that were randomly selected. Specifically, we used PCR primers binding outside the genomic region complementary to the HDR, and the amplicons were then digested with HaeIII. We confirmed the presence of HDR-mediated mutations in the 6 positive clones, as the HaeIII cut site was present in all of them (Appendix A). Clone 6 did not show a 304 bp band corresponding to an undigested amplicon, but only a strong 152 bp band corresponding to the digestion products, suggesting that both alleles were HDR-corrected for this clone, but not for the other ones (Appendix A).

The 14 amplicons analyzed for the presence of the HaeIII cut site were also processed for MiSeq sequencing. All cellular clones had two *TRIM5* alleles of approximately equal abundance (≈50%), with the exception of clone 38 that seemingly had 3 alleles. Of the six positive clones after screening, five showed monoallelic editing, including the insertion of both R332G and R335G mutations as well as the formation of a HaeIII cut site, as expected (Figure 1). In each of these clones, a second *TRIM5* allele had no mutation at all. In these monoallelically modified clones, only four of the eight intended mutations were found. None of the HDR-modified alleles had the three desired mutations upstream of the HaeIII-creating mutation, and all of them had an unexpected substitution in the PAM, G→T instead of the intended G→C (both substitutions are silent). In addition to the modified and unmodified alleles, clone 38 had an allele of unclear genesis that contained a large deletion (Figure 1). Two of the 8 negative clones randomly selected (31 and 33) had a single nucleotide deletion at position -2 from the cut site (Appendix A). Clone 6 was the only clone to have both *TRIM5* alleles edited by HDR. One of its alleles had five of the intended eight mutations (R332G, R335G, HaeIII and PAM sites). The other one also had five mutations, including R332G, but it lacked the R335G mutation (Figure 1).

### 3.2. HIV-1 Restriction Activity in TRIM5-Edited Clones

In order to determine permissiveness to HIV-1 of the gene-edited clones, as well as TRIM5α stimulation by IFN-I, cells were infected in the absence or presence of IFN-β with increasing amounts of VSV-G-pseudotyped HIV-1_NL-GFP_, a GFP-expressing, propagation-incompetent ΔEnv/ΔNef version of the HIV-1 clone NL4-3 [65]. The percentage of infected cells was determined by analyzing GFP expression using flow cytometry. The titration curves were remarkably grouped for the 7 screen-negative clones included in this experiment, suggesting an absence of strong clone-to-clone variation in permissiveness to lentiviral transduction in Jurkat cells. In the absence of IFN-β, only clone 6 showed a significant decrease (≈10-fold) in permissiveness to HIV-1_NL-GFP_, compared with the rest of the clonal populations (Figure 2). In the presence of IFN-β, all cell populations showed decreased permissiveness to HIV-1_NL-GFP_, indicating that one or more ISGs inhibited HIV-1, as expected [12]. IFN-I treatment resulted in significant restriction of HIV-1_NL-GFP_ in three clones (8, 12 and 30) that had monoallelic HDR editing, compared to screen-negative clones (≈5-fold), and restriction was strikingly strong in clone 6 cells (≈40-fold) (Figure 2). Clone 17 could not be included in this experiment, but we found it to behave similarly to clone 30 in a separate infection experiment (Appendix A).

We then analyzed HIV-1 restriction in clones 6 and 8 side-by-side with Jurkat cells stably transduced with a retroviral vector expressing R332G-R335G huTRIM5α [53]. In the absence of IFN-β, HIV-1_NL-GFP_ was restricted in both clone 6 and in the retrovirally transduced cells expressing R332G-R335G huTRIM5α, compared with the parental WT cells (Figure 3). In the presence of IFN-β, clone 6 showed a 50-fold reduction in permissiveness to infection compared to parental cells, and the monoallelically HDR-modified clone 8 also showed a modest restriction effect. In contrast, HIV-1 restriction in the cells retrovirally transduced with R332G-R335G TRIM5α was not stimulated by IFN-β (Figure 3), consistent with the transgene being expressed from a non-IFN-I-inducible promoter.

### 3.3. Knockdown Validation of TRIM5α Antiviral Function

A cellular clone might show reduced permissiveness to HIV-1 due to an IFN-I-inducible factor other than TRIM5α. In order to ensure that the low permissiveness of clone 6 to HIV-1 was due to expression of the edited TRIM5α, we knocked down TRIM5α in these cells as well as in the unedited parental (WT) cells. The shRNA used here binds to a different TRIM5α region than the one targeted for mutagenesis. Untransduced cells were eliminated by antibiotic selection. Two monoallelically *TRIM5*-edited clonal populations, clone 8 and clone 12, were also included in this experiment. Cells were infected with increasing doses of HIV-1_NL-GFP_ in the presence or absence of IFN-β. We found that knocking down TRIM5α rescued HIV-1_NL-GFP_ infection of clone 6 cells, both in the absence and the presence of IFN-β, though the effect was stronger in the presence of IFN-β, as expected (Figure 4). In contrast, TRIM5α knockdown had no effect on the infection of the unedited parental cells by HIV-1_NL-GFP_. The control shRNA used, which targets the nonhuman gene luciferase, did not affect permissiveness to HIV-1. Knocking down TRIM5α also increased HIV-1_NL-GFP_ infection of clones 8 and 12, which have only one R332G-R335G *TRIM5* allele, but the effect was seen only in the presence of IFN-β and was smaller compared to clone 6. In summary, TRIM5α knockdown-mediated rescue of HIV-1 infectivity in clones 6, 8 and 12 correlated with the extent of HIV-1 restriction as seen in Figure 2, and also correlated with the enhancement of restriction by IFN-β. Thus, decreased permissiveness to HIV-1 in clones 6, 8 and 12 was due in large part to the antiviral effect of edited *TRIM5* alleles in these clones. R332G-R335G huTRIM5α retains the ability to restrict the N-tropic strains of murine leukemia virus (MLV), as shown previously [53], whereas B-tropic MLV is not restricted by either WT or mutated huTRIM5α. Thus, cells were also infected with N-MLV_GFP_ and B-MLV_GFP_ in order to assess TRIM5α restriction capabilities on a target other than HIV-1. N-MLV_GFP_ infection was strongly increased by knocking down TRIM5α in all cell populations and in the absence or presence of IFN-β, indicating that all cell populations expressed restriction-competent TRIM5α alleles (Figure 4). B-MLV_GFP_ infection of parental cells as well as clone 8 and clone 12 cells was slightly increased by knocking down TRIM5α, and only in the presence of IFN-β. This suggests with WT huTRIM5α, but not the R332G-R335G mutant, has weak inhibitory activity against B-MLV, which is revealed by the IFN-I treatment. Thus, *TRIM5* editing to allow HIV-1 targeting had little or no impact on its capacity to restrict a different retrovirus.

### 3.4. Mechanism of Inhibition by Edited TRIM5α

TRIM5α-mediated restriction decreases retroviral cDNA synthesis by reverse transcription, and this mechanism of restriction is counteracted by the proteasome-inhibiting drug MG132 [39,66]. qPCR was used to analyze HIV-1 cDNA synthesis in clone 6 and in unmodified parental cells in the presence or absence of IFN-β and MG132 (Figure 5). Following a short exposure to HIV-1_NL-GFP_, total cellular DNAs were prepared, and qPCR was performed using primers complementary to the HIV-1 vector-specific gene GFP, which is representative of late reverse transcription cDNA. To adjust for DNA amounts in the reactions, GAPDH DNA was quantified as well. Consistent with the known mechanism of action for TRIM5α, we found HIV-1 cDNA amounts to be significantly smaller (≈6-fold) in clone 6 cells, as compared to the parental cells, in the absence of drug (Figure 5). In the presence of IFN-β, this effect was more pronounced, consistent with the stimulation of TRIM5α-mediated restriction by IFN-I. Conversely, the reduction in HIV-1 cDNA amounts in clone 6 was only 2.5-fold compared to WT in the presence of MG132, showing that MG132 attenuated this restriction phenotype, as expected. Accordingly, MG132 rescued HIV-1 cDNA synthesis by a larger magnitude in clone 6 (4.1-fold) than in the WT cells (1.7-fold), in the absence of IFN-β. Similar findings were observed in the presence of IFN-β (Figure 5). In conclusion, these results show that HIV-1 infection is inhibited via the expected mechanism in the *TRIM5*-edited clone 6 cells.

## 4. Discussion

Despite an increased focus from the HIV-1 research community on cure research in recent years [67], aiming in particular at eliminating latent reservoirs [3], we are still far from a sterilizing or functional cure. Similarly, and despite some recent conceptual advances such as novel strategies to elicit the production of broadly neutralizing antibodies [68], vaccine prospects remain seemingly out of reach [69,70]. The field of HIV-1 drug research continues to show impressive progress, such as the ongoing emergence of long-acting antiviral drugs [71] and of the HIV-1 capsid protein as a novel drug target [72]. However, curative treatments are still elusive, and multidrug resistance is a persistent concern [73]. Thus, the search for alternative strategies is still a priority. HIV-1 is one of the rare infectious diseases for which a genetic intervention is sensible. Indeed, it is a life-long infection whose treatment is expensive and which results in diminished life expectancy and lessened quality of life, even in successfully treated patients [74,75,76]. If successful, a one-off genetic intervention would represent an attractive option for HIV-1 patients. Most genetic strategies presently explored with the aim of inhibiting HIV-1 involve the knockdown or knockout of the HIV-1 co-receptor CCR5, but this approach is not without caveats (see introduction).

In this study, we demonstrate that editing of the intrinsic, innate effector TRIM5α protects human cells against infection by HIV-1. Although such an approach has been conceptualized before by us and others [60,77], this study constitutes the first proof-of-concept for the protection against a pathogen provided by editing of an innate effector, to the best of our knowledge. Yet, we encountered a major difficulty in that biallelic modification seemed to be a rare event, with only one out of 6 HDR-edited clones having the desired mutations on both alleles. Clones with R332G-R335G introduced into only one of the two alleles showed a mild restriction phenotype, which necessitated IFN-I treatment to be revealed. Co-expression of restrictive and nonrestrictive TRIM5α alleles is expected to lead to a weak restriction phenotype. Indeed, distinct TRIM5α proteins can form heterodimers [57,58], and a TRIM5α lattice containing both capsid-targeting and -nontargeting monomers would probably bind the viral core weakly. If TRIM5α is to be pursued in gene editing approaches to suppress HIV-1, it is crucial to develop strategies that improve the rate of biallelic editing. When this project was initiated, co-transfection of CRISPR components and donor DNA was the only option available for precise genome editing. Methods for introducing discrete substitutions in eukaryotic genomes have vastly improved and diversified since and now include adeno-associated viral vector-mediated delivery of the donor DNA for HDR [78]; various pharmacological, physical or genetic methods to improve HDR rates [79,80,81,82]; and base editing, which does not require DNA cuts nor a donor DNA [83]. Taking advantage of these recent innovations, we consider it likely that it will soon be possible to achieve efficient biallelic editing of *TRIM5*, along with other restriction factor genes, in human cells, resulting in a profound disruption of HIV-1 infectivity.

## Figures and Tables

**Figure 1 viruses-13-00024-f001:**
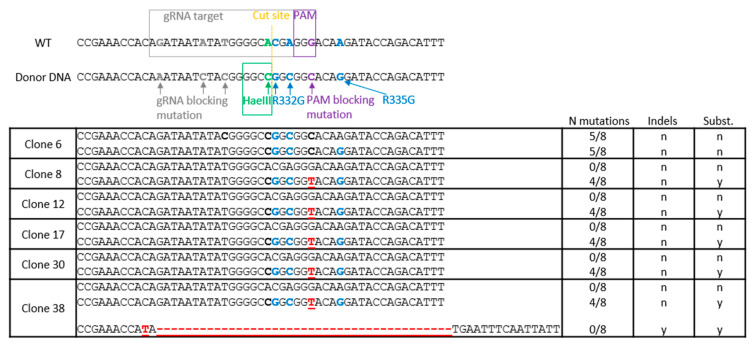
Tripartite-motif-containing protein 5 (*TRIM5)* editing strategy and outcome. Top, the *TRIM5* region targeted for mutagenesis is aligned with the reverse-complement sequence of the donor single-stranded donor DNA (ssDNA) central region. The mutated nucleotides are shown in colors, and their purpose is indicated. Substitutions leading to R332G and R335G mutations are in blue. The PAM-blocking silent mutation is in purple. The silent mutation creating an HaeIII restriction cut site is in green. Additional silent mutations in the gRNA-binding region are in grey. Bottom, sequence of *TRIM5* alleles found in clonal Jurkat populations following electroporation of clustered regularly interspaced short palindromic repeats (CRISPR)-Cas9 ribonucleoproteins (RNPs) and the ssDNA and PCR-based screening. For each cell population, alleles were found in equal amounts, except for clone 38, in which the relative amounts were 44% (unedited parental (WT) allele), 24% (homology-directed repair (HDR)-edited allele) and 32% (indel-containing allele). Substituted nucleotides are in bold. Substitutions leading to R332G and R335G mutations are in blue. Indels or undesirable substitutions are in red. The number of desired mutations for each allele is shown on the right, along with the presence (y) or absence (n) of insertions and deletions.

**Figure 2 viruses-13-00024-f002:**
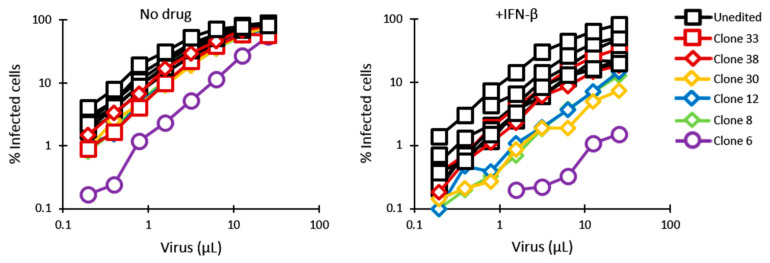
Permissiveness of HDR-edited and unedited control Jurkat cell populations to infection with HIV-1_NL-GFP_. Cells were infected with increasing amounts of HIV-1_NL-GFP_ in the absence (**left**) or presence (**right**) of IFN-β (10 ng/mL), and the % of GFP-positive cells was determined 2 days later by FACS. Results are presented for 5 clonal populations containing HDR-edited alleles (diamond symbols and purple circles) and 7 randomly chosen negative clones (squares). Clones containing a deletion in one of the alleles are shown in red.

**Figure 3 viruses-13-00024-f003:**
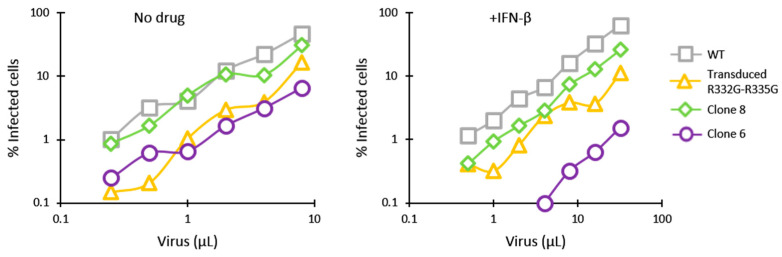
IFN-β treatment stimulates HIV-1 restriction by mutated endogenous TRIM5α but has no effect on retrovirally transduced TRIM5α. Parental Jurkat cells, clone 6 and clone 8 cells, and cells stably transduced with R332G-R335G were infected with increasing amounts of HIV-1_NL-GFP_ in the absence or presence of IFN-β, and the % of GFP-positive cells was determined 2 days later by FACS.

**Figure 4 viruses-13-00024-f004:**
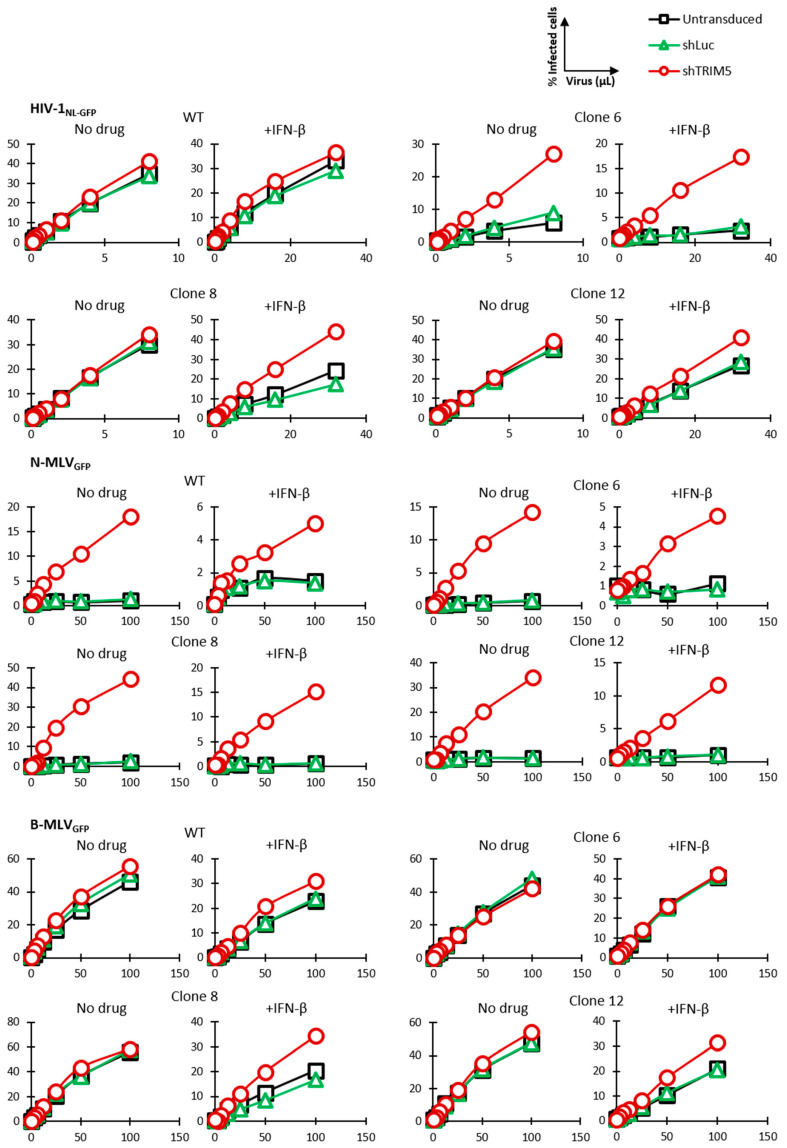
TRIM5α knockdown rescues HIV-1 from restriction in gene-edited cells. Parental Jurkat cells and clone 6, 8 and 12 cells were lentivirally transduced with shRNAs targeting TRIM5α or Luc as control or left untransduced. Cells were then infected with increasing amounts of HIV-1_NL-GFP_, N-MLV_GFP_ or B-MLV_GFP_ in the absence or presence of IFN-β. The% of GFP-positive cells was determined 2 days later by FACS.

**Figure 5 viruses-13-00024-f005:**
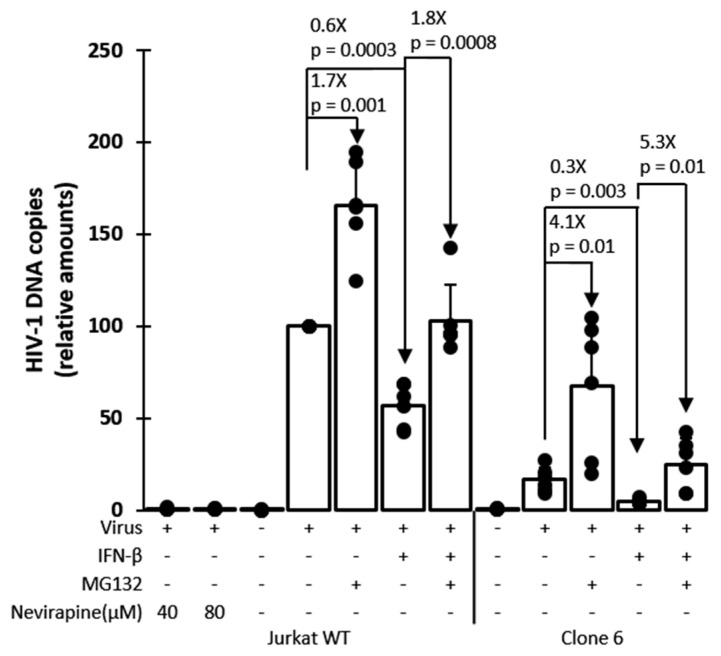
HIV-1 cDNA synthesis is inhibited in *TRIM5*-edited clone 6 Jurkat cells but is rescued by MG132 treatment. Parental (WT) Jurkat cells and clone 6 cells were treated or not as indicated with IFN-β, MG132 or the reverse transcriptase inhibitor nevirapine (as a DNA contamination control). Cells were then infected for 6 h with HIV-1_NL-GFP_, followed by DNA extraction and qPCR with primers specific for HIV-1 vector cDNA (GFP sequence) and for the cellular gene GAPDH for normalization purposes. Results are presented as HIV-1 cDNA copy numbers adjusted according to GAPDH copy numbers and normalized to the no-drug WT cells control, which is set at 100%. *p*-values were calculated using the Student’s *t*-test.

## Data Availability

The data presented in this study are openly available at FigShare: https://figshare.com/projects/D_saulniers_et_al_2021/93836.

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
