# Peer review of "Editing of the TRIM5 Gene Decreases the Permissiveness of Human T Lymphocytic Cells to HIV-1"

_viruses, 2020, doi:10.3390/v13010024_

Round 1

Reviewer 1 Report

Editing of the TRIM5 gene decreases the permissiveness of human T lymphocytic cells to HIV-1 by Kevin Désaulniers et al

Comments to the Editor-in-Chief  (Vaccines, MDPI)

My Decision is Accept after minnor revision

This is a brillant study from a molecular biology wiew point, these data open a clinical perspective of their findings in HIV-1 seropositive patients, including those with supressed viral load (less than 50 copies).

-All relevant information has been include in the abstract and the organization of the manuscript is perfect. The discussion is really clear and the explanation of methods is exhaustive. However, some smaill requirements are neccesary before its final acceptation.
Thus, my Decision is Minnor revision

Although such an approach has been conceptualized before by others and these authors also [60,77], this study is the first proof-of-concept for the protection against a pathogen provided by editing of an innate effector.

These authors have generated  huTRIM5α SPRY domain mutant libraries that were screened for their capacity to restrict HIV-1. When over-expressed through retroviral transduction, R332G-R335G huTRIM5αinhibits the spread of HIV-1 by 20- to 40-fold and provides a survival advantage compared to untransduced cells [53,55].

Previously, they transfected plasmids encoding CRISPR components (Cas9 and guide RNA, gRNA) into HEK293T cells, along with a donor DNA for HDR that bear the desired mutations [59], and obtained several  cell clones containing corrected alleles [60]. They havee electroporated CRISPR ribonucleoprotein (RNP) complexes and a mutation donor DNA into Jurkat T cells in order to  introduce the R332G and R335G mutations.

The present approach is relevant since introducing the desired  mutations in TRIM5 by gene editing represents an alternative, as the therapeutic gene will  be expressed at physiological levels as well as in a IFN-I-dependent fashion. These approach are relevant at the molecular level. They reported that successfully TRIM5-edited cells have decreased permissiveness to HIV-1, and that restriction is stimulated by IFN-I as expected

The biallelic modification seemed to be a rare event, with only one out of 6 HDR edited clones affecting both alleles

Minnor comments

Introduction

Please, explain better this sentence…. ¨and are inefficient in some anatomical sanctuaries [4]¨

Explain the meaning of palindromic repated…..line 40-41 for a general audience in viral and immunology field, spetially for clinicalreseachers also.

I am not tottaly agree with you since HIV_1 tropism for CXCR4 is 1 % of infected patients.  These author indicate ..¨However, results from the ‘Essen patient’ demonstrate that this approach alone will not be successful for patients that harbor even very small amounts of HIV that can use the CXCR4 co-receptor for entry into cells [10,11].¨ This CXCR4 tropism is really 1 % of HIV-1 infected patients ori t is really lower as comapare to CCR5 tropism in seropositive patients. Please, revise and confirm your sentence about CXCR4. Shall you verify, please this sentence

Line 56-57. Please, explain the meaning of SVY in this sentence….¨ At its C-terminus, TRIM5α has a SPRY domain whose sequence determines retroviral target specificity [28,29].¨

Line 64-66.. ¨ Restriction by TRIM5α is species-specific, cell-type-specific and virus-specific [43-45]. Although some HIV-1 variants are significantly restricted by human TRIM5α (huTRIM5α), in particular in HLA:B27+ or B57+ elite controller patients [24,46,47]. Please, add details about the involved mechanism.

-These authors textually indicates…¨Zinc finger nucleases (ZFNs) have been designed to knock out the CCR5 gene, and phase I clinical trials demonstrated that patient-derived ZFN engineered T cells were effective in controlling HIV viral loads in some patients following treatment interruption [6].¨ Please, add the reason here.

-Describe the follow process how has been calculate the percentage of GFP-positive cells was determined on a FC500 MPL cytometer (Beckman Coulter, CA) with the FCS Express 6 analysis software (De novo software, CA). Please, explain better clinical details.

Methods

Line 98..Extend the information of the follows sentence…¨ Cell lines were obtained from J. Luban (University of Massachusetts School of Medicine). Jurkat¨

Line 99. Explain why the medium has been switched to RPMI at the time of virus production (HEK293T cells).

All culture media were supplemented with 10% fetal bovine serum (FBS), penicillin/streptomycin  (HyClone) and Plasmocin (InvivoGen, San Diego, CA). The Jurkat cells stably transduced with a lentivirus over-expressing R332G-R335G TRIM5α were described before [53]. Please, explain why it is neccesary plasmocin or FBS addition here? Is the proliferaiton affected by FBS in these cells lines?

Line 10.4. ¨ The Jurkat cells stably transduced with a lentivirus over-expressing R332G-R335G TRIM5α were described before [53].¨  Briefly, describe it this molecular procedure for a clinical audience also.

Line 106. Explain the procedure for a general and clinical audience also. The CRISPR gRNA was designed using the Zhang Lab algorithm available online at crispr.mit.edu.

Line 110. Which is the meaning of by IDT? Include it.

Line 111. Explain with more detail the system IDT ALT® CRISPR-Cas9 system to edit TRIM5.

Line 112. ¨ For each well, we prepared 1 μL of tracrRNA:crRNA:Cas9 RNP complex containing 22 pmol of crRNA:tracrRNA duplex and 18 pmol of Cas9 enzyme. Explain the

Line 115. …¨ We used the Neon transfection system (Thermo Fisher Scientific) with the following parameters: 1323 V, 10 ms  and 3 pulses. The plates were incubated at 37°C, 5% CO2 for 48 h.¨. Shall you explain why you selected these physical conditions here?

Line. 156…¨ Infections of Jurkat cells were performed in 96-well plates seeded at 10,000 cells per well the day before. Where applicable, treatment with IFN-β (PeproTech, Rocky Hill, NJ) at a final concentration  of 10 ng/ml was initiated 16 h prior to infection¨. Have you published previous data with these concentrations? In such case, include the reference or another published data.

Line. 161-162. PLease, indicate some picture that describe this sentence..

The percentage of GFP-positive cells was determined on a FC500 MPL cytometer (Beckman Coulter, CA) with the FCS Express 6 analysis software (De novo software, CA).

Line 169. Why the exactly number of 750,000 cells per well were seeded were seeded here?

Lin 171…Is there any tipo error here….…. ¨and in a….¨

Line 184. PLease, briefly describe the ΔCt methor for a clincial audience also.

Line 185. Explain the process of normalization. ¨ Relative HIV-1 cDNA copy numbers were then normalized to levels found in infected/untreated control (parental) cells which were set at 100%.¨

Line 243. ¨ The percentage of infected cells was determined by analyzing GFP expression using flow cytometry.¨. Please, indicate some representative flow cytomertry photo here.

Line 396. Repeted twice… blood-2010-09-309591 [pii]

References

Please, check all references since some doi… appaers in the next line (doi…..?)

Major comments

These authours indicate that …¨Previously, these authors  transfected plasmids encoding CRISPR components (Cas9 and guide RNA, gRNA) into HEK293T cells, along with a donor DNA for HDR that bear the desired mutations [59], and obtained several  cell clones containing corrected alleles [60]. However, no antiviral effect was observed, which was due to several possible reasons, including the fact that in this cell line TRIM5 bears an additional mutation that might inactivate its antiviral properties [60].¨

Explain these reasons of the lack of effect/s here.

Author Response

Dec 18, 2020

Dear Editor,

I am addressing here the comments raised by Reviewer 1 of our manuscript, "Editing of the TRIM5 gene decreases the permissiveness of human T lymphocytic cells to HIV 1", for publication as a review article in Viruses. I thank the reviewer for his/her encouraging comments. I believe that the revised version is greatly improved as a result.

Below I am inserting our point-by-point response to the comments:

This is a brillant study from a molecular biology wiew point, these data open a clinical perspective of their findings in HIV-1 seropositive patients (…). The discussion is really clear and the explanation of methods is exhaustive. However, some smaill requirements are neccesary before its final acceptation. Thus, my Decision is Minnor revision Minnor comments

Thank you for this appreciation.

Introduction: Please, explain better this sentence…. ¨and are inefficient in some anatomical sanctuaries [4]¨

Anatomical sanctuaries are tissues where HIV-1 may infect some cells without being easily targeted by immune cells or by drugs. To make it more clear, I have added an example of sanctuary for HIV-1: the central nervous system.

Explain the meaning of palindromic repated…..line 40-41 for a general audience in viral and immunology field, spetially for clinical reseachers also.

The word “palindromic” is present here only to explain the acronym “CRISPR”. It is probably not worth it to detail to the reader the molecular mechanism of the CRISPR system in the organisms in which it was discovered (bacteria), which is what we would have to do if we were to explain what is palindromic in this system. To make it more clear that we are simply explaining the acronym, I used capital letters for the first letter of each word.

I am not tottaly agree with you since HIV_1 tropism for CXCR4 is 1 % of infected patients.  These author indicate ..¨However, results from the ‘Essen patient’ demonstrate that this approach alone will not be successful for patients that harbor even very small amounts of HIV that can use the CXCR4 co-receptor for entry into cells [10,11].¨ This CXCR4 tropism is really 1 % of HIV-1 infected patients ori t is really lower as comapare to CCR5 tropism in seropositive patients. Please, revise and confirm your sentence about CXCR4. Shall you verify, please this sentence

We agree that CCR5 is the main co-receptor and that the successful knockout of CCR5 would probably protect a significant number of patients against HIV-1. However, CXCR4-tropic strains may become predominant in some of these patients, as demonstrated by this published case. In any case, we have precised and toned down our statement: “However, results from a patient in whom CXCR4-tropic HIV-1 rebounded following transplantation with CCR5d32/d32 cells suggest that this approach alone may not be successful…”

Line 56-57. Please, explain the meaning of SVY in this sentence….¨ At its C-terminus, TRIM5α has a SPRY domain whose sequence determines retroviral target specificity [28,29].¨

SPRY is not an acronym. We have added the other name under which this domain is known, B30.2. We have also added a few words to better explain that this domain determines which viruses are inhibited by the protein.

Line 64-66.. ¨ Restriction by TRIM5α is species-specific, cell-type-specific and virus-specific [43-45]. Although some HIV-1 variants are significantly restricted by human TRIM5α (huTRIM5α), in particular in HLA:B27+ or B57+ elite controller patients [24,46,47]. Please, add details about the involved mechanism.

We have added an explanatory sentence: “because evolutionary pressure from cytotoxic T lymphocytes in these patients results in the emergence of capsid mutants that increase sensitivity to TRIM5α”.

-These authors textually indicates…¨Zinc finger nucleases (ZFNs) have been designed to knock out the CCR5 gene, and phase I clinical trials demonstrated that patient-derived ZFN engineered T cells were effective in controlling HIV viral loads in some patients following treatment interruption [6].¨ Please, add the reason here.

We have added the mention that these cells are CCR5-knockout.

-Describe the follow process how has been calculate the percentage of GFP-positive cells was determined on a FC500 MPL cytometer (Beckman Coulter, CA) with the FCS Express 6 analysis software (De novo software, CA). Please, explain better clinical details.

We modified the text to be more explanatory: “The percentage of GFP-positive cells was determined using a FC500 MPL cytometer (Beckman Coulter, CA) with the FCS Express 6 analysis software (De novo software, CA). The percentage of GFP+ cells among total intact cells was determined by gating them after excluding cell debris.”

Line 98..Extend the information of the follows sentence…¨ Cell lines were obtained from J. Luban (University of Massachusetts School of Medicine). Jurkat¨

We have deleted this sentence, as these cells have been in our lab for 15 years and have been used in many previously published articles. They are not newly introduced materials that would deserve detailed description of origin.

Line 99. Explain why the medium has been switched to RPMI at the time of virus production (HEK293T cells).

This is because we are using the virus to infect cells that grow in RPMI, not DMEM. We have added this clarification. This is standard procedure in our lab.

All culture media were supplemented with 10% fetal bovine serum (FBS), penicillin/streptomycin  (HyClone) and Plasmocin (InvivoGen, San Diego, CA). The Jurkat cells stably transduced with a lentivirus over-expressing R332G-R335G TRIM5α were described before [53]. Please, explain why it is neccesary plasmocin or FBS addition here? Is the proliferaiton affected by FBS in these cells lines?

Most mammalian cells require 10% serum to grow well, and we use low-concentration plasmocin to prevent contamination by mycoplasma.

Line 10.4. ¨ The Jurkat cells stably transduced with a lentivirus over-expressing R332G-R335G TRIM5α were described before [53].¨  Briefly, describe it this molecular procedure for a clinical audience also.

We have added a line of explanation: “Briefly, these cells were exposed to an HIV-1 vector containing both the huTRIM5α mutant and the gene of resistance to puromycin, followed by puromycin treatment to kill untransduced cells.”

Line 106. Explain the procedure for a general and clinical audience also. The CRISPR gRNA was designed using the Zhang Lab algorithm available online at crispr.mit.edu.

We have made this section more explanatory: “The CRISPR gRNA target within TRIM5 (5’AGATAATATATGGGGCACGA) was selected using the Zhang Lab algorithm available online at crispr.mit.edu [60]. We used the Integrated DNA Technologies (IDT, Kanata, ON, Canada) ALT® CRISPR-Cas9 RNP system to edit TRIM5. The CRISPR RNA (crRNA) and transactivating CRISPR RNA (tracrRNA), which are the RNA components of the RNP complex, were synthesized by IDT. The previously described single-stranded donor DNA (ssDNA) that includes mutations encoding 332G and 335G as well as silent mutations [60] was also synthesized by IDT.”

Line 110. Which is the meaning of by IDT? Include it.

It is explained (see above).

Line 111. Explain with more detail the system IDT ALT® CRISPR-Cas9 system to edit TRIM5.

Details have been added (see above).

Line 112. ¨ For each well, we prepared 1 μL of tracrRNA:crRNA:Cas9 RNP complex containing 22 pmol of crRNA:tracrRNA duplex and 18 pmol of Cas9 enzyme. Explain the

We have removed the mention of a “well”, which was irrelevant. The whole genome editing paragraph of the Methods section has been improved.

Line 115. …¨ We used the Neon transfection system (Thermo Fisher Scientific) with the following parameters: 1323 V, 10 ms  and 3 pulses. The plates were incubated at 37°C, 5% CO2 for 48 h.¨. Shall you explain why you selected these physical conditions here?

This is based partly on recommendations by the manufacturer, and partly on our internal optimization experiments. There is no previous reference; in fact, this article will serve as reference for future studies.

Line. 156…¨ Infections of Jurkat cells were performed in 96-well plates seeded at 10,000 cells per well the day before. Where applicable, treatment with IFN-β (PeproTech, Rocky Hill, NJ) at a final concentration  of 10 ng/ml was initiated 16 h prior to infection¨. Have you published previous data with these concentrations? In such case, include the reference or another published data.

A reference has been added (Dufour et al., 2018)

Line. 161-162. PLease, indicate some picture that describe this sentence.. The percentage of GFP-positive cells was determined on a FC500 MPL cytometer (Beckman Coulter, CA) with the FCS Express 6 analysis software (De novo software, CA).

The FACS method is now described in more details: “The percentage of GFP-positive cells was determined using a FC500 MPL cytometer (Beckman Coulter, CA) with the FCS Express 6 analysis software (De novo software, CA). The percentage of GFP+ cells among total intact cells was determined by gating them after excluding cell debris”. We sometimes add methods schematics in order to explain novel or complex protocols, but this is not the case here; in this study, the one method that truly needs to be explained is the HDR-mediated repair, and we do this in Fig. 1. The rest is very standard.

Line 169. Why the exactly number of 750,000 cells per well were seeded were seeded here?

It is not based on any particular protocol. The student who was doing these experiments just found that it was the right amount of cells. It is within the range of cell concentrations that we use in this well format.

Lin 171…Is there any tipo error here….…. ¨and in a….¨

Indeed, it is now corrected, thank you.

Line 184. PLease, briefly describe the ΔCt methor for a clincial audience also.

We have added the following clarification for what ΔCt is: “(cycle threshold for sequence of interest minus cycle threshold for the control sequence)”. Going into more details would not be appropriate in this paper, as ΔCt is a basic molecular biology method for which abundant information is available.

Line 185. Explain the process of normalization. ¨ Relative HIV-1 cDNA copy numbers were then normalized to levels found in infected/untreated control (parental) cells which were set at 100%.¨

I am not sure how to explain it further. If you look at Fig. 5, you will see that the control for these experiments (+HIV, no drug) are all with a value of 100. This is because all the values were normalized to the value obtained for the control. We did this for each repeat of the experiment and then did the statistics that are shown on the figure.

Line 243. ¨ The percentage of infected cells was determined by analyzing GFP expression using flow cytometry.¨. Please, indicate some representative flow cytomertry photo here.

I am writing this on Dec 19. We have less than two weeks to submit the new version of the manuscript. Our university is on lockdown due to COVID-19 until Jan 5, and even I cannot access my office. Thus, I do not have access to the raw FACS data at this moment. However, we have started uploading all primary data to a public repository (https://figshare.com/projects/D_saulniers_et_al_2021/93836). All the primary FACS files will be uploaded to FigShare in the first week of January.

Line 396. Repeted twice… blood-2010-09-309591 [pii]

We fixed this problem.

Please, check all references since some doi… appaers in the next line (doi…..?)

All DOIs have been deleted from the reference list.

Major comments

These authours indicate that …¨Previously, these authors  transfected plasmids encoding CRISPR components (Cas9 and guide RNA, gRNA) into HEK293T cells, along with a donor DNA for HDR that bear the desired mutations [59], and obtained several  cell clones containing corrected alleles [60]. However, no antiviral effect was observed, which was due to several possible reasons, including the fact that in this cell line TRIM5 bears an additional mutation that might inactivate its antiviral properties [60].¨ Explain these reasons of the lack of effect/s here.

In the previous paper about this project (Dufour et al., 2018), we indeed found that endogenous TRIM5α was poorly restrictive in HEK293T cells, and the antiviral activity was marginally increased by interferon treatment, suggesting that low expression levels were not responsible for low restriction. Looking at available HEK293T TRIM5 sequences (http://hek293genome.org/v2/), we noticed that TRIM5 alleles in this cell lines present dozens of SNPs compared with reference sequences. Thus, it is possible that one or several mutations in HEK293T cells reduce TRIM5α restriction activity. We cannot propose a better explanation, and we have added the link to the website cited above.

We have made some additional minor corrections that had escaped my attention at the first submission. We have also added the link to the data repository to which we have uploaded all raw data pertaining to this paper.

Sincerely,

Dr. Lionel Berthoux

Laboratory of antiviral immunity                                                    

Department of Medical Biology

3351, boulevard des Forges

C.P. 500

Trois-Rivières (Québec)

G9A 5H7

Phone: 819-376-5011 x4466

Fax: 819-376-5084

[email protected]

Reviewer 2 Report

Summary

Researchers modified endogenous huTRIM5a to restrict HIV-1 infection in human T lymphocytic cells by creating mutations R332G/R335G to the TRIM5 gene through CRISPR Cas-9 mediated homology-directed repair. Generally, the idea is great, and the research provides evidence to better understand TRIM5 inhibitory function on HIV infection.

Specific comments:

  1. Overall clearer labelling of x- and y-axis for all graphs needed. Data analysis reports are not easy to understand for readers, the graphs look so messy, please make the figure clear to read, especially Fig. 2, 3, and 4. Virus measured in concentration or MOI instead of volume for repeatability.
  2. Can authors provide the repeats for all analysis data to show clear error bar and p value?
  3. Figure 2, Why does clone 6 start at 1 uL virus in +IFN-b condition? Authors should include all data they have done, please provide reasonable explanation for the data not shown in figures.
  4. Figure 3, it would be helpful to confirm the importance of two residue mutations in HIV restriction if adding transduced TRIM5α-wt, R332G, or R335G as controls.
  5. More concise language throughout would be beneficial.

Author Response

Dec 18, 2020

Dear Editor,

I am addressing here the comments raised by Reviewer 2 of our manuscript, "Editing of the TRIM5 gene decreases the permissiveness of human T lymphocytic cells to HIV 1", for publication as a review article in Viruses. I thank the reviewer for his/her encouraging comments. I believe that the revised version is greatly improved as a result.

Below I am inserting our point-by-point response to the comments:

Researchers modified endogenous huTRIM5a to restrict HIV-1 infection in human T lymphocytic cells by creating mutations R332G/R335G to the TRIM5 gene through CRISPR Cas-9 mediated homology-directed repair. Generally, the idea is great, and the research provides evidence to better understand TRIM5 inhibitory function on HIV infection.

Thank you for this appreciation.

Specific comments:

  1. Overall clearer labelling of x- and y-axis for all graphs needed. Data analysis reports are not easy to understand for readers, the graphs look so messy, please make the figure clear to read, especially Fig. 2, 3, and 4. Virus measured in concentration or MOI instead of volume for repeatability.

For Fig. 2 and Fig. 3, we have added the axis titles on the axes. For Fig. 4, we left it as it was. There are a lot of graphs in Fig. 4 and it would somewhat overload this figure to have the axis titles for each graph. These are infection experiments similar to the ones in Fig. 2 and Fig. 3, thus, we believe that the reader will have no problem understanding it (and of course, the legend is still present at the top of the figure).

Virus concentrations expressed in MOIs are useful when comparing different viruses. In the experiments for this manuscript, we are comparing infectivity of a given virus in different cell populations. In this context, MOI values do not add information. In all our previous articles for this project, virus doses were similarly expressed in volumes (Pham, Gene Ther, 2010, PMID 20619429; Veillette, Retrovirology, 2013, PMID 23448277; Jung, Hum Gene Ther, 2015, PMID 26076730; Dufour, PLOS One, 2018, PMID 29373607). Other groups in the TRIM5 field also routinely use volume rather than MOI when the latter is not needed; examples of this can be found in the articles cited in response to comment #2. This being said, the MOI can be easily deduced from the infectivity curves shown. For instance, in Fig. 2, 1 μL of the HIV-1 vector has an MOI of ≈0.1 on unedited cells (about 10% infected cells).

However, we have added the following in the Methods section, which helps understanding the way infections are done in these experiments: “The virus amounts used were determined so that we would obtain > 0.1% infected cells (FACS analysis threshold in the conditions used) while staying below saturation concentrations (≈ 50% infected cells), based on preliminary titration experiments.” We have also removed the mentions of titrations done in CRFK cells, as these titrations were indeed done but not used in the study.

We thank the reviewer, as having a close look at this issue made us realize that we hadn’t specified what MOI had been used in the qPCR experiment shown Fig. 5. It was 0.1 and it is now mentioned in the Methods section.

  1. Can authors provide the repeats for all analysis data to show clear error bar and p value?

We have modified Fig. 5 so that all the values are now apparent instead of just showing the bars and standard deviations. The infection experiments shown in Figs 2-4 were not done in triplicates. Typically, such infection experiments are repeated 3-4 times if done at a single virus dose. When performing titrations with multiple virus doses as shown here, they are not. Our previously published papers reflect this, but I would like to also point out that the same is true for the most high-profile teams in the TRIM5 field, including the Sodroski group (ex. PMID 21680743), the Emerman group (17038183), The Bieniasz group (23505372), the Diaz-Griffero group (24314652), etc.

  1. Figure 2, Why does clone 6 start at 1 uL virus in +IFN-b condition? Authors should include all data they have done, please provide reasonable explanation for the data not shown in figures.

This is because in these cells, infections with less than 1 μL of virus resulted in less than 0.1% infected cells, which is below the threshold for robustness in this assay, and thus we didn’t use the data. This is specified in the sentence now added to the Methods section (see response to comment #1).

  1. Figure 3, it would be helpful to confirm the importance of two residue mutations in HIV restriction if adding transduced TRIM5α-wt, R332G, or R335G as controls.

We didn’t want to revisit this point, as it has been shown several times already that the double mutant is more restrictive than either single mutation alone (Pham et al., 2010, 2013; Veillette et al., 2013). The double mutant huTRIM5α transduced cells only come here as controls for the trim5-edited cells.

  1. More concise language throughout would be beneficial.

We have made a number of changes in the manuscript to make it easier for the reader; the other reviewer had many comments requesting clarifications and language edits, which helped achieving this.

We have made some additional minor corrections that had escaped our attention at the first submission. We have also added the link to the data repository to which we have uploaded all raw data pertaining to this paper.

Sincerely,

Dr. Lionel Berthoux

Laboratory of antiviral immunity                                                    

Department of Medical Biology

3351, boulevard des Forges

C.P. 500

Trois-Rivières (Québec)

G9A 5H7

Phone: 819-376-5011 x4466

Fax: 819-376-5084

[email protected]

Round 2

Reviewer 2 Report

The authors properly addressed my concerns.